# Empowering Teachers with Enhanced Knowledge via Variable Scale Distillation Framework

## Abstract

Knowledge distillation, a widely used model compression technique, enables a smaller student network to replicate the performance of a larger teacher network by transferring knowledge, typically in the form of softened class probabilities or feature representations. However, current approaches often fail to maximize the teacher's feature extraction capabilities, as they treat the semantic information transfer between teacher and student as equal. This paper presents a novel framework that addresses this limitation by enhancing the teacher's learning process through the Variable Scale Distillation Framework. Central to our approach is the Rescale Block, which preserves scale consistency during hierarchical distillation, allowing the teacher to extract richer, more informative features. In extensive experiments on the CIFAR100 dataset, our method consistently outperforms state-of-the-art distillation techniques, achieving an average accuracy improvement of 2.12%. This demonstrates the effectiveness of our approach in fully leveraging the teacher's capacity to guide the student, pushing the boundaries of knowledge distillation.

## 1 Introduction

Knowledge distillation, introduced by Hinton et al. (2015), revolutionized model compression by showing how knowledge from a large teacher network can be distilled into a smaller student network. This process typically involves training a teacher network with extensive parameters and transferring its soft logits to guide the training of the student network. This technique is commonly used in various downstream fields, including object detection Chen et al. (2017); De Rijk et al. (2022), segmentation Liu et al. (2019), etc. Previous work has proposed multiple approaches, including distilling knowledge via feature-based methods Romero et al. (2014); Ahn et al. (2019); Chen et al. (2022); Guo et al. (2023); Tian et al. (2019), logit-based methods Chen et al. (2020a); Hinton et al. (2015); Zheng et al. (2020); Mirzadeh et al. (2020), and relation-based methods Huang et al. (2022); Li et al. (2022); Park et al. (2019); Peng et al. (2019). Further advancements have integrated self-supervised learning into the knowledge distillation framework Xu et al. (2020); Yang et al. (2021); Lee et al. (2020), leveraging pretext tasks to enhance the distillation process and improve model performance.

However, in previous work, both networks receive identical input, neglecting the teacher's stronger feature extraction and under-utilizing its complex architecture and larger parameter set. To address this limitation, our work presents a novel approach inspired by image super-resolution Ledig et al. (2017). It leverages the teacher network's superior feature extraction capability by presenting it with more detailed, fine-grained image input. Unlike previous methods, we propose Variable Scale Distillation Framework, where the teacher network is trained on images of varying resolutions to capture both high-level semantic features and finer details. This methodology allows the teacher model to fully exploit its larger capacity without altering the original semantic content of the images.

Furthermore, we propose a Rescale Block that ensures scale consistency during hierarchical distillation. This is critical in maintaining the alignment of feature maps between teacher and student networks, addressing the common issue of feature mismatch when applying Variable Scale Distillation Framework. Different from most previous methods, our method incorporates multi-task training,

combining self-supervised learning with image classification. It enables the teacher model to aggregate knowledge from both tasks and impart it effectively to the student model. Our contributions are listed as follows:

- **Variable Scale Distillation Framework**: In this novel framework, we enable the teacher network to learn more fine-grained semantic features from high-resolution images, allowing the student model to absorb richer knowledge.

- **Rescale Block**: By maintaining scale consistency between teacher and student feature maps, we solve the feature mismatch problem, ensuring the effectiveness of knowledge transfer.

- **State-of-the-art Performance**: Our method consistently surpasses existing state-of-the-art approaches in tasks such as image classification, few-shot learning, and linear classification, showcasing its generalizability to other downstream applications.

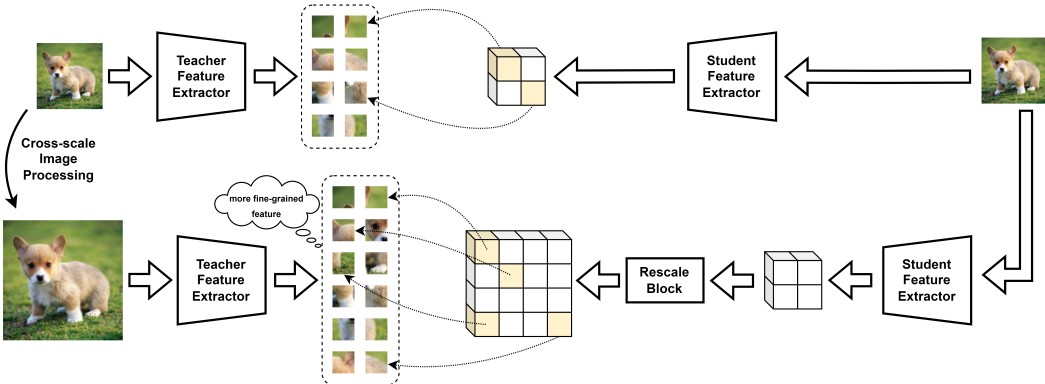

Figure 1: In the Variable Scale Distillation Framework, the teacher network can learn more fine-grained feature representations. When training the student network, the feature map processed by the Rescale Block can learn the teacher network's dark knowledge more comprehensively.

## 2 RELATED WORK

### 2.1 KNOWLEDGE DISTILLATION

Knowledge distillation trains a smaller network using the supervision signals from both ground truth labels and a larger network Gou et al. (2021). The seminal concept of knowledge distillation was introduced by Hinton et al. Hinton et al. (2015), which achieved knowledge transfer through soft probability distributions. When comparing our method with common state-of-the-art knowledge distillation methods, FitNet Romero et al. (2014) proposes a method where the student network learns the intermediate features of the teacher network to achieve better performance. AT Zagoruyko & Komodakis (2016b) improves the performance of a student CNN network by forcing it to mimic the attention maps of a powerful teacher network. RKD Park et al. (2019) transfers mutual relations of data examples to the student network. CRD Tian et al. (2019) trains the student network to capture significantly more information in the teacher's representation of the data. SSKD Xu et al. (2020) demonstrates that the seemingly different self-supervision task in knowledge distillation can serve as a simple yet powerful solution. HSAKD Yang et al. (2021) proposes appending several auxiliary classifiers to hierarchical intermediate feature maps to generate diverse self-supervised knowledge and perform the one-to-one transfer to thoroughly teach the student network. To summarize, while these methods have advanced knowledge distillation by enhancing the transfer of knowledge, they often overlook the potential for leveraging diverse input transformations. In response to these limitations, our work enables the teacher network to provide richer, more comprehensive guidance to the student network.

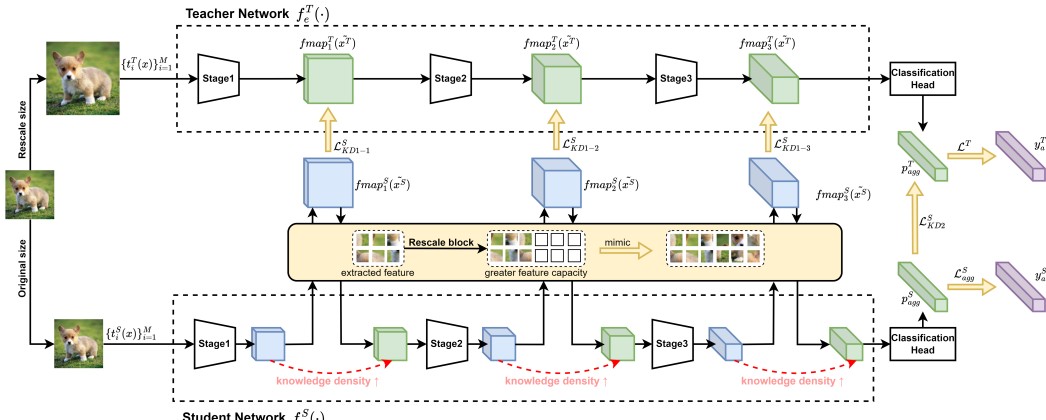

Figure 2: An overview of our method. Mathematical symbols appearing in the figure will be declared in this chapter later. The teacher network receives images in variable scales and then is trained through the constraints of $\mathcal{L}^{\mathcal{T}}$. The student network receives the original size images and is equipped with Rescale Blocks to ensure scale consistency of hierarchical knowledge distillation. $\mathcal{L}^{S}_{KD1-1}$, $\mathcal{L}^{S}_{KD1-2}$, $\mathcal{L}^{S}_{KD1-3}$, $\mathcal{L}^{S}_{KD2}$, $\mathcal{L}^{S}_{agg}$ jointly constrain the training of student networks.

## 2.2 SELF-SUPERVISED LEARNING

Self-supervised learning is a paradigm for learning visual features from large-scale unlabeled data Jing & Tian (2020); Liu et al. (2021). It allows us to train deep neural networks without artificial labels, providing useful representations for specific downstream tasks. For the pretext tasks of self-supervised learning, many effective methods have been proposed in SimCLR Chen et al. (2020b), including the restoration of the original image Pathak et al. (2016); Zhang et al. (2016); Kolesnikov et al. (2019); Noroozi & Favaro (2016); Gidaris et al. (2018b), and the classification of original categories of processed images Dosovitskiy et al. (2016); Gidaris et al. (2018a); Misra et al. (2016), among others. The color permutation and rotation in Lee et al. (2020) feed the network with augmented images and force it to recognize the rotation angle and channel sorting order. During the training of the student network, it has to understand the semantic information contained in the image, thereby achieving better performance. While previous methods have significantly advanced self-supervised learning to extract rich semantic features, our work integrates self-supervised tasks directly into the knowledge distillation process, forming an aggregated task that combines both classification and self-supervision. This allows for an end-to-end training framework, improving student's performance by leveraging the full potential of the teacher's guidance.

## 3 METHOD

In this section, we detail our Variable Scale Distillation Framework. An overview of the proposed approach is illustrated in Figure 2. During the teacher network's training, we apply a self-supervised technique to process the input images. This enables the teacher to capture more fine-grained semantic features. For the student model, we also employ self-supervision to enhance its ability to learn semantic information from the images. Additionally, the student's training is guided by both the teacher network's logits and the parameters from the intermediate feature layers.

### 3.1 TEACHER NETWORK

#### 3.1.1 VARIABLE SCALE IMAGE REFINEMENT

In most existing knowledge distillation methods, the teacher and student networks typically receive identical inputs. However, these methods fail to fully exploit the teacher network's intricate architecture and extensive parameters. To overcome this, we employ the variable scale image refinement technique aimed at harnessing the teacher network's advanced feature extraction capabilities.

As discussed in Section 4.1, we experimented with several common input processing techniques (Figure 3) to enhance the teacher network's input by increasing the image size and enriching its semantic content. Based on our findings, we selected bilinear interpolation to double the width and height of the input image, which serves as the core of our Variable Scale Distillation Framework.

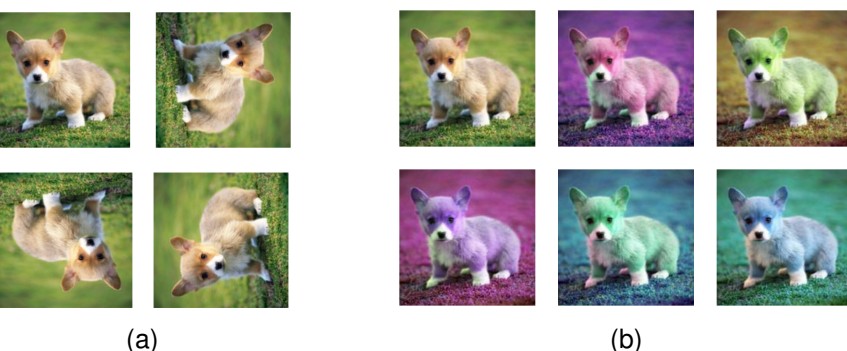

(a)                             (b)

Figure 3: Pretext tasks applied to the input of both teacher network and student network. (a) shows rotation-based augmentation. (b) shows channel-permutation-based augmentation.

### 3.1.2 Aggregated-Task Learning with Self-supervision

To enhance the teacher and student networks' ability to generalize, we introduce an additional self-supervised task that broadens the dimensions of the final vector space during training. This self-supervised transformation preserves the semantic information of the input images, helping the network better capture the underlying structure. To reduce training costs and fully leverage the existing label information in the dataset, we perform training in an aggregated vector space that combines both image classification and self-supervised tasks. This joint training approach allows the model to infer as effectively as an aggregated model after a single training process.

Let the network function be represented as $f(\cdot)$, which can be decomposed into a feature extractor $f_e(\cdot, \omega_e)$ and a linear classification head $f_c(\cdot, \omega_c)$, where $\omega$ represents the weight tensors. Consider $x \in \mathcal{X}$ as a mini-batch from the training dataset $\mathcal{X}$.

We define a total of M various self-supervised transformations $\{t_i\}_{i=1}^M$ within the transformation space $\mathcal{M} = \{1, ..., M\}$, and N classification labels within the label space $\mathcal{N} = \{1, ..., N\}$. The aggregated vector space for the task is $\mathcal{A} = \mathcal{M} \otimes \mathcal{N} = \{(1,1), ..., (M, N)\}$, where $\otimes$ means the cartesian product, and the dimension of $\mathcal{A}$ is $||A|| = M \times N$.

By applying transformations on $x$, we obtain $\tilde{x} = \{t_i(x)\}_{i=1}^M$. The feature extractor generates the feature embedding vector $\tilde{z} = f_e(\tilde{x}, \omega_e) \in \mathbb{R}^d$, where $d$ is the embedding size. The predicted distribution in the aggregated vector space $\mathcal{A}$ is $p_{agg} = \sigma(f_c(\tilde{z}, \omega_c), \tau) \in \mathbb{R}^{M \times N}$, while the standard class probability distribution is $p_{cls} = \sigma(f_c(z, \omega_c), \tau) \in \mathbb{R}^N$, where $\tau$ is the temperature hyperparameter to scale the smoothness of distribution.

### 3.1.3 Training

Considering that the teacher network needs to be trained on the aggregated vector space, we use $\tilde{x^T} = \{t_i^T(x)\}_{i=1}^M$ as the input instead of $x$, where the superscript $T$ indicates that $x$ is the input of the teacher network. Therefore, the loss function that constrains the convergence of teacher network training is expressed as Eq.(1), where $y_a^T$ means the ground-truth labels of $\tilde{x^T} = \{t_i^T(x)\}_{i=1}^M$ in aggregated space $\mathcal{A}$. $\mathcal{L}_{CE}$ represents cross entropy loss.

$$\mathcal{L}^T = \frac{1}{M} \sum_{i=1}^M \mathcal{L}_{CE}(\sigma(f_c^T(\tilde{z^T}, \omega_c), \tau), y_a^T). \tag{1}$$

## 3.2 STUDENT NETWORK

### 3.2.1 HIERARCHICAL KNOWLEDGE DISTILLATION

During inference, feature maps at different levels capture various semantic features. Lower-level feature maps typically focus on fine-grained details, such as texture, while higher-level feature maps represent more abstract, global information. To effectively utilize the hierarchical feature maps from the teacher network, we employ a Hierarchical KD Branch to transfer the knowledge from multiple layers of the teacher. This allows the student network to learn not only from the final soft targets but also from the intermediate representations that are critical in the teacher's evaluation process.

To further refine the feature representations for guiding the student model, one transformation in the pretext task is doubling the input image's width and height. However, this creates a mismatch in input sizes between the teacher and student networks, posing a challenge for hierarchical distillation. To address this, we designed a Rescale Block that ensures scale consistency between the feature maps during the distillation process. This module also captures the hierarchical feature maps in the aggregated task training, allowing the student to learn from both detailed and high-level features.

### 3.2.2 TRAINING

We denote $\tilde{x^S} = \{t_i^S(x)\}_{i=1}^M$ as the input of the student model. Then we decompose the total loss of training the student model into two parts: the loss between predictions and ground-truth labels under the aggregated task, and the mimicry loss between the hierarchical feature maps and final logits of the teacher and student models, respectively, during the knowledge distillation.

**Loss of aggregated task.** Given that the student network and the teacher network follow the same rules and are trained under the aggregation task without considering knowledge distillation, we define the form of this loss function to be similar to Eq.(1). At the same time, we introduce the binary cross entropy loss between the intermediate feature maps and the ground-true classification as Eq.(3).

$$\mathcal{L}_{agg1}^S = \frac{1}{M} \sum_{i=1}^M \mathcal{L}_{CE}(\sigma(f_c^S(\tilde{z^S}, \omega_c), \tau), y_a^S), \tag{2}$$

$$\mathcal{L}_{agg2}^S = \frac{1}{M} \sum_{i=1}^M \sum_{j=1}^K \mathcal{L}_{CE}(\sigma(fmap_j^T(\tilde{x^T}), \tau), y_a^S). \tag{3}$$

**Loss of knowledge distillation.** In the context of knowledge distillation loss, it can be further decomposed into the hierarchical feature map loss $L_{KD1}^S$ and the final logits loss $L_{KD2}^S$. For the former, we assume that we simultaneously sample $K$ intermediate layers in both the teacher network and student network for knowledge distillation. These are denoted as $fmap_{k=1}^{T/S}(x^{\tilde{T}/S})$, This is quantitatively measured using KL divergence $D_{KL}$, with the loss formulated as in Eq.(4). In the same form, we define the loss of the latter as Eq.(5).

$$\mathcal{L}_{KD1}^S = \frac{1}{M} \sum_{i=1}^M \sum_{j=1}^K D_{KL}(\sigma(fmap_j^T(\tilde{x^T}), \tau) \| \sigma(fmap_j^S(\tilde{x^S}), \tau)), \tag{4}$$

$$\mathcal{L}_{KD2}^S = \frac{1}{M} \sum_{i=1}^M \sum_{j=1}^K D_{KL}(p_{agg}^T \| p_{agg}^S). \tag{5}$$

We combine the loss of aggregated task and loss of knowledge distillation to obtain the overall loss $L^S$ of the student model training as Eq.(6), where $\lambda_1$ and $\lambda_2$ are the hyper-parameters used to balance different losses.

$$\mathcal{L}^S = \lambda_1 \mathcal{L}_{KD1}^S + \lambda_2 \mathcal{L}_{KD2}^S + \lambda_3 \mathcal{L}_{agg1}^S + \lambda_4 \mathcal{L}_{agg2}^S. \tag{6}$$

# 4 EXPERIMENTS

Our experiments are divided into three key components. First, in Section 4.1, we present an ablation study to evaluate the effects of various pretext tasks and image processing techniques in our framework. Next, in Section 4.2, we provide a comprehensive comparison with state-of-the-art methods, demonstrating the efficacy of our proposed approach. Finally, in Section 4.3, we conduct an in-depth analysis of the proposed method. CIFAR100 Krizhevsky (2009) serves as the primary dataset, with STL10 Coates et al. (2011) and TinyImageNet used for linear classification experiments in Section 4.3. We employ several network architectures as backbones, including ResNet He et al. (2016), WideResNet Zagoruyko & Komodakis (2016a), VGG Simonyan & Zisserman (2015), ShuffleNet Zhang et al. (2018), and MobileNet Howard et al. (2017).

## 4.1 ABLATION STUDY

In this ablation study, we systematically investigate the impact of different transformations, loss functions, and training pipelines on the student network's performance. Our primary focus is on evaluating the effects of our Variable Scale Distillation Framework, designed to exploit the superior feature extraction capabilities of the teacher network. In this setup, the teacher network is WRN40-2, and the student network is WRN40-1. As our framework specifically targets the teacher network, we experiment with various processing techniques for the teacher, while maintaining a consistent transformation (rotation) for the student model.

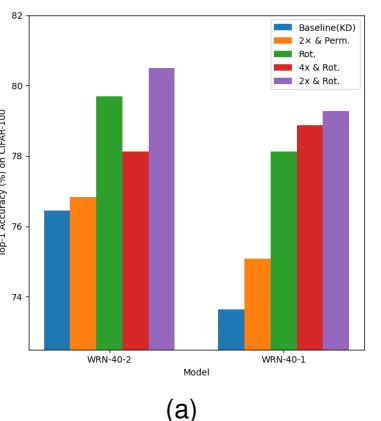
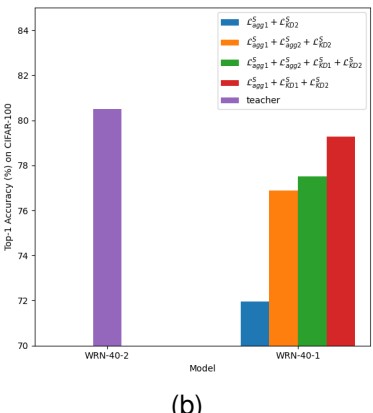

(a)                     (b)

Figure 4: Comparison of Top-1 accuracy(%) of student model and teacher model under different transformations (a) and different loss functions (b).

**Different transformations.** As illustrated in Figure 4(a), the introduction of various pretext tasks $\tilde{x^T}$ positively influences the performance of the student network to varying extents. Notably, increasing the input size for the teacher network enhances the performance of the student network as well. In the figure, $4\times$ indicates that the length and width of the original input image are quadrupled using linear interpolation. The label **Rot.** signifies the use of rotation as a self-supervised task within the aggregated learning framework, while **Perm.** Denotes the permutation of image channels in different orders.

The results demonstrate that employing rotation as a pretext task yields superior accuracy compared to channel permutation. Furthermore, when utilizing the rotation pretext task, the performance of the Variable Scale Distillation Framework is significantly better with a twofold enlargement compared to a fourfold enlargement of the input size.

**Different loss functions.** Following the student model loss function defined in Eq.(6), we observed that, aside from the classification loss function $\mathcal{L}_{agg1}^S$ and the distillation loss $\mathcal{L}_{KD2}^S$, the remaining components proved to be extraneous. Consequently, we conducted an ablation study focusing on the effectiveness of various loss functions during the training of the student model.

As depicted in Figure 4(b), we selected the combination of loss functions $\mathcal{L}_{KD1}^{S} + \mathcal{L}_{KD2}^{S} + \mathcal{L}_{agg1}^{S}$ as the unified loss function for our approach. This selection is underpinned by the observation that it yields the highest accuracy for the student model, thereby reinforcing the effectiveness of the Variable Scale Distillation Framework.

**Different pipelines.** To validate the efficacy of the proposed Variable Scale Distillation Framework in enhancing the performance of various distillation techniques, we incorporated our framework into SSKD, training multiple teacher-student pairs.

As illustrated in Table 1, the application of the Variable Scale Distillation Framework resulted in significant improvements in the accuracy of the student model across different configurations. This finding underscores that our innovative approach not only allows the teacher network to extract more fine-grained dark knowledge but also enables the student model to acquire more effective information that enhances its classification capabilities.

Table 1: Ablation experiments of the Variable Scale Distillation Framework on SSKD.

| Teacher
Student | WRN40-2
WRN40-1 | WRN40-2
WRN16-2 | ResNet32x4
ResNet 8x4 |
|---|---|---|---|
| SSKD | 76.04 | 76.13 | 76.20 |
| SSKD + our framework | 76.66 | 77.54 | 80.21 |
| $\Delta$ | 0.62 ↑ | 1.41 ↑ | 4.01 ↑ |

**Correlation matrix of logits.** To further investigate the mechanisms by which our proposed method enhances the classification performance of the student model, we visualized the correlation matrix of logits from a selected teacher-student pair in Figure 5. The results reveal that the integration of the Variable Scale Distillation Framework facilitates the transfer of finer-grained dark knowledge to the student model. Consequently, the correlation of logits between the teacher and student models across different categories is notably reduced compared to baseline methods, as evidenced by the lighter hues (Figure 5(b)) in the visual representation.

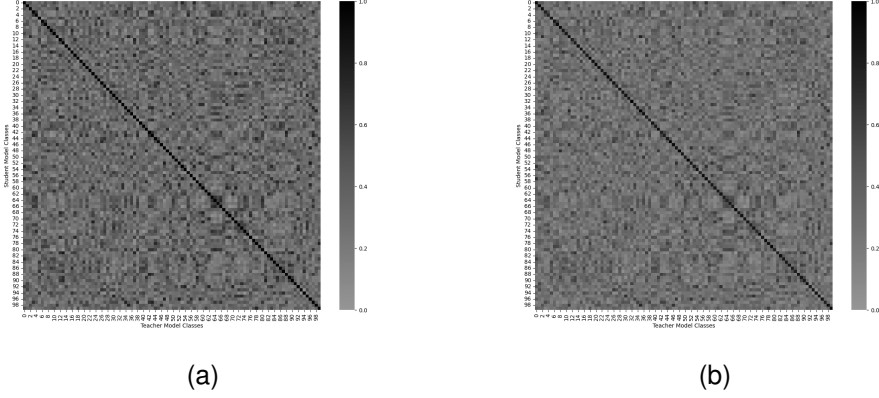

(a)                                             (b)

Figure 5: Visualization of the correlation matrix of logits from teacher-student pair without our method (the left one with darker color), and with our method (the right one with lighter color).

## 4.2 COMPARISON WITH STATE-OF-THE-ART METHODS

In this section, we present a comprehensive comparison of our proposed method against contemporary state-of-the-art (SOTA) approaches across ten distinct teacher-student pairs. The experimental results are organized into two tables, delineating scenarios where the teacher and student networks share similar architectures (Table 2) versus those with different architectures (Table 3).

Table 2: Top-1 accuracy (%) on CIFAR100 through different SOTA methods of knowledge distillation between similar architectures. The numbers in **Bold** and underline denote the best and the second-best results.

| Teacher
Student | WRN40-2
WRN16-2 | WRN40-2
WRN40-1 | ResNet56
ResNet20 | ResNet32×4
ResNet8×4 | VGG13
VGG8 |
|---|---|---|---|---|---|
| Without KD | 73.64 | 72.24 | 69.63 | 72.51 | 70.68 |
| KD | 74.92 | 73.54 | 70.66 | 73.33 | 72.98 |
| FitNet | 75.75 | 74.12 | 71.60 | 74.31 | 73.54 |
| AT | 75.28 | 74.45 | 71.78 | 74.26 | 73.62 |
| RKD | 75.40 | 73.87 | 71.48 | 74.47 | 73.72 |
| CRD | 76.04 | 75.52 | 71.68 | 75.90 | 74.06 |
| SSKD | 76.04 | 76.13 | 71.49 | 76.20 | 75.33 |
| HSAKD | 78.67 | 78.12 | 73.73 | 77.69 | 75.93 |
| ours | **81.82** | **79.27** | **77.81** | **83.28** | **79.35** |

Table 3: Top-1 accuracy (%) on CIFAR100 through different SOTA methods of knowledge distillation between different architectures. The numbers in **Bold** and underline denote the best and the second-best results.

| Teacher
Student | VGG13
MobileNetV2 | ResNet50
MobileNetV2 | ResNet32×4
ShuffleNetV1 | ResNet32×4
ShuffleNetV2 | WRN-40-2
ShufflNetV1 |
|---|---|---|---|---|---|
| Without KD | 65.79 | 65.79 | 70.77 | 73.12 | 70.77 |
| KD | 67.37 | 67.35 | 74.07 | 74.45 | 74.83 |
| FitNet | 68.58 | 68.54 | 74.82 | 75.11 | 75.55 |
| AT | 69.34 | 69.28 | 74.76 | 75.30 | 75.61 |
| RKD | 68.50 | 68.46 | 74.20 | 75.74 | 75.45 |
| CRD | 68.49 | 70.32 | 75.46 | 75.72 | 75.96 |
| SSKD | 71.53 | 72.57 | 78.44 | 78.61 | 77.40 |
| HSAKD | **79.27** | **79.43** | 78.39 | 80.86 | 80.11 |
| ours | 76.28 | 77.94 | **81.55** | **82.49** | **83.69** |

The second partition of each table details the accuracy of the student network when trained independently, without the influence of the teacher network. In contrast, the third partition showcases the top-1 accuracy of the student network when various knowledge distillation methods are employed.

Remarkably, across all five teacher-student pairs with similar architectures, our method consistently outperforms existing techniques, achieving superior accuracy. Notably, in the ResNet32x4 & ResNet8x4 configuration, the accuracy of the student network trained via our approach surpasses that of the best-competing methods by a substantial margin of 5.59%. This improvement highlights the efficacy of our Variable Scale Distillation Framework innovations in facilitating more effective knowledge transfer from teacher to student, ultimately enhancing the student's performance.

## 4.3 FURTHER ANALYSIS

**Efficacy under the few-shot scenario.** We assess the effectiveness of our method by comparing it with established approaches such as KD, CRD, SSKD, and HSAKD in few-shot scenarios, retaining 25%, 50%, and 75% of the training samples, as summarized in Table 4. Notably, in the 25% few-shot scenario, the student model trained using our methodology achieves an accuracy of 70.50%. This performance is particularly striking when contrasted with the conventional KD method, which attains only a marginally higher accuracy of 70.66% in the 100% data training scenario. This discrepancy can be attributed to our incorporation of rotation as a pretext task, which enriches the dataset in few-shot situations. Furthermore, the features extracted from the teacher network in our framework provide enhanced guidance for the student network's learning process, distinguishing our approach from other methods.

Table 4: Top-1 accuracy (%) comparison on CIFAR100 in few-shot scenario with various ratio of training samples. The values in the table are the accuracy of the student network when ResNet56 and ResNet20 are used as teacher-student pairs.

| few-shot ratio | 25% | 50% | 75% | 100% |
|---|---|---|---|---|
| KD | 65.15 | 68.61 | 70.34 | 70.66 |
| CRD | 65.80 | 69.91 | 70.98 | 71.68 |
| SSKD | 67.82 | 70.08 | 70.18 | 71.49 |
| HSAKD | 68.50 | 72.18 | 73.26 | 73.73 |
| ours | **70.50** | **74.55** | **76.28** | **77.81** |

Table 5: Linear classification top-1 accuracy (%) of transfer learning on STL-10 and TinyImageNet. We use WRN40-2 as the teacher network and ShuffleNetV1 as the student network.

| | CIFAR100 $\rightarrow$ STL10 | CIFAR100 $\rightarrow$ TinyImageNet |
|---|---|---|
| KD | 73.25 | 32.05 |
| FitNet | 73.77 | 33.28 |
| AT | 73.47 | 33.75 |
| CRD | 74.44 | 34.30 |
| SSKD | 74.74 | 34.54 |
| HSAKD | 75.62 | 38.65 |
| ours | **77.91** | **47.22** |

**Linear classification.** We further explored the generalization capabilities of the student model's feature extractor. To this end, we froze the encoder of the student model trained on CIFAR100 and subsequently trained two classification heads on STL10 and TinyImageNet, respectively. The results of this evaluation are presented in Table 5. Remarkably, the student networks trained via our method outperformed all other mainstream approaches. This outcome suggests that our method equips the network with knowledge that transcends the specific images within the training dataset. The integration of Variable Scale Distillation Framework and self-supervised tasks facilitates a focus on more salient semantic features during the learning process, thereby enabling the network to achieve superior generalization performance across diverse datasets.

## 4.4 LIMITATIONS

In this study, we focused on specific pretext tasks such as image rotation and channel permutation to improve the generalization ability of the student model. That said, we did not explore other potentially effective self-supervised tasks, such as random masking, which could present another promising avenue for enhancing the feature extraction capabilities of both the teacher and student models. The primary reason is to maintain computational efficiency and simplicity in our experimentation pipeline, as the training of models with random masking typically requires additional resources and longer convergence times due to the need for feature recovery mechanisms.

We acknowledge that incorporating random masking could potentially further improve the student's ability to learn fine-grained and contextually rich representations from the teacher. Future work could explore the combination of our framework with random masking to assess its effectiveness in distillation scenarios.

## 5 CONCLUSION

In this study, we propose the Variable Scale Distillation Framework, and integrating it with self-supervised tasks to create a unified framework. Furthermore, to maintain size consistency among feature maps during hierarchical distillation, we present the Rescale Block, which facilitates effective feature map alignment. This innovative approach empowers student networks to acquire richer and more generalized semantic feature representations, ultimately achieving state-of-the-art performance on standard image classification benchmarks within the realm of knowledge distillation.

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
