# OpenReview forum: "Empowering Teachers with Enhanced Knowledge via Variable Scale Distillation Framework"
_ICLR.cc/2025/Conference — Submitted to ICLR 2025_

### Official Review · Reviewer_JDMz · 2024-11-02

**Soundness:** 1
**Presentation:** 2
**Contribution:** 1
**Rating:** 5
**Confidence:** 4

**Summary:**

This paper proposes the Variable Scale Distillation Framework, which incorporates several techniques. First, it trains the teacher model on images doubled in size using cross-entropy loss, then distills the enhanced features into the student model with original-sized images to fully leverage the teacher's intricate architecture and extensive parameters. During the distillation process, to match the feature map sizes between multiple layers of the teacher and student networks, the paper introduces a Rescale Block, which increases the student model's feature map size by copying and pasting existing features. Additionally, to incorporate the benefits of supervised techniques, it rotates the input images and broadens the dimensions of the final vector space while training both the teacher and student models. Finally, binary cross-entropy loss is applied between the intermediate feature maps and the ground-truth labels. To evaluate the performance of the proposed method, they compare it with state-of-the-art methods in supervised, few-shot, and transfer learning tasks across various datasets.

**Strengths:**

This paper explores various techniques to enhance the performance of the student model.

**Weaknesses:**

The motivation for and effects of the proposed method are overall unclear. The paper claims that doubling the image size as input to the teacher model helps fully leverage the teacher network’s intricate architecture and extensive parameters, which other existing KD methods overlook. However, the authors provide neither explanations nor experimental evidence for this assertion. Could you provide comparison results for the teacher model's feature representations when using doubled-size images versus original-size images as input? Additionally, could you explain why using quadrupled-size images decreases the performance of both the teacher and student models?

Additionally, the authors apply some self-supervised techniques but do not clarify which advantages of self-supervised learning were intended in this paper. and this paper does not demonstrate the effectiveness of training the teacher model and distilling knowledge to the student, even at the expense of increased training cost. Could you provide experimental results on directly training the student model using the proposed self-supervision technique for the teacher model, Variable Scale Image Refinement, while keeping the teacher model fixed (no training) and leaving the rest of the proposed method unchanged?

The paper claims to introduce a Hierarchical KD Branch to distill feature knowledge from multiple teacher layers to the student model, but this approach is already common in feature-based knowledge distillation (e.g., FitNet [Romero et al., 2015], ReviewKD [Chen et al., 2021]). Furthermore, the concrete principles behind the Rescale Block method and $fmap_j$ are not adequately explained.

The experimental section also lacks sufficient evidence to support the authors' claims. The paper only compares a single pair of student and teacher networks in both the ablation study and further analysis sections, limiting the reliability of the results.

**Questions:**

1. Could you explain the reason why doubling the image size as input to the teacher model helps fully leverage the teacher network’s intricate architecture and extensive parameters?
2. Did you only use rotation and resizing transformations for the pre-processing of the input images (M=2) during teacher training?"
3. Could you explain why using quadrupled images reduces the performance of both the teacher and student models?
4. Could you cite the paper that proposes the supervision technique for broadening the dimensions of the final vector space during training?
5.Could you explain what information is transferred from the teacher through aggregated-task learning with self-supervision compared to existing KD methods, and how this benefits the student model?
6.  Could you provide more detailed information about  Rescale Block and $fmap_j$?
7.  Did you utilize additional linear classification layers for Eq. (3) and Eq. (4) to match the feature map size and labels?
8.  Could you provide experimental results on directly training the student model using the proposed self-supervision technique for the teacher model, Variable Scale Image Refinement, while keeping the teacher model fixed (no training) and leaving the rest of the proposed method unchanged?
9.  Could you report the ablation study and further analysis with more pairs of teacher and student networks (both similar and different architectures)?
10. Could you report the results of the teacher model trained with your proposed methods, as well as without them?
11.In Section 4.1.4, "Correlation Matrix of Logits" (page 7), isn't a higher correlation between the teacher's logits and the student's logits preferable?
12. Could you report the ablation study on the method without training the teacher?
13. Could you provide comparison results for the teacher model's feature representations when using doubled-size images versus original-size images as input?

Things to improve the paper that did not impact the score:
1. Please follow the formatting instructions of ICLR regarding citations within the text, ensuring that \citep{} and \citet{} are used appropriately.
2. Possible typo: In line 3 of the Knowledge Distillation section (2.1) on page 2, 'Hinton et al.' is written twice.
3. Please unify the notation for referencing, deciding whether to use 'Equation 5, Figure 5' or 'Eq. (5), Fig. (5).'
4. Possible typo: On page 5, "$\lambda_1$ and $\lambda_2$" should be corrected to "$\lambda_1$ ,$\lambda_2$ , $\lambda_3$  and $\lambda_4$ "

---

> ### Author Response · Authors · 2024-11-21
> **Rebuttal for reviewer JDMz**
>
> Thank you for reviewing our work and providing detailed comments and suggestions on work motivation, experiments, writing, etc. Your suggestions are a great motivation for us to improve the article. Thank you again for your recognition of our work and your comments.
> # Q1: Why variable scale input improves model performance
> From a qualitative analysis, the teacher model has a larger framework and more parameters, and has more learning capacity than the student model. So we propose to train the teacher with images with higher scale variation to make full use of the teacher's learning ability.
>
> From the experimental results, our proposed method can achieve sota performance in the field of knowledge distillation. In addition, in Figure 5 of our article, we try to use visualization to explain why variable scale input can improve model performance. When using variable scale input, the correlation between the teacher model logits and the student model logits of different categories is lower, which is reflected in the lighter color of the visualization image. The results show that compared with the original scale input method, this method has less reference learning of wrong categories during the student learning process from the teacher.
>
> # Q2: Use rotation and variable scale for teacher training
> Yes. When training the teacher network, we use variable scale input and rotation to provide the network with more learning content. Variable scale input provides more pixels on the same image for the teacher to learn from, and rotation provides more samples for the teacher model. Both methods match the larger knowledge capacity of the teacher model.
>
> # Q3: Quadrupled images reduce the performance
> Thank you for pointing this out. The teacher model and student model obtained with quadruple-sized input are still improved compared to the baseline, but there is a certain performance degradation compared to double-sized input. According to our analysis from the perspective of the teacher model, a reasonable and intuitive explanation is that quadruple-sized input adds too many pixels for the teacher to learn, which exceeds the teacher model's learning capacity. Of course, this explanation may be very intuitive, so we will add a visualization of the correlation coefficient matrix of quadruple input in subsequent versions. We believe that visualization can provide a reasonable explanation for the performance degradation phenomenon.
>
> # Q4: Cite a paper
> Of course, [1] aggregates the self-supervision task and the image classification task during training, thereby expanding the dimension of the final vector space. This article is also cited in our work.
>
> # Q5-1: Information transferred through aggregated-task learning
> Under the aggregated task, the teacher model transfers the logits in the aggregation label. Taking rotation as a self-supervised task as an example, the logits not only contain information about the category of the input image, but also indicate how many degrees the image has been rotated.
>
> This is beneficial to the student model in the following ways. First, the introduction of self-supervised tasks such as rotation is to increase the training samples and thus better utilize the learning ability of the teacher network. The teacher can transfer richer dark knowledge through knowledge distillation. Second, aggregating the self-supervised task with the classification task can, on the one hand, obtain a simpler end-to-end training framework, and on the other hand, prevent the destruction of the knowledge learned by the network in two separate trainings, which has been theoretically demonstrated and experimentally proved in [1].
>
> # Q5-2: Details about Rescale Block and $famp_j$
> Of course. We may not have made you understand the Resacale Block and $famp_j$ well in the text, and we will add some explanations in the final version.
>
> Rescale Block: When we input the rescaled image into the teacher model, the scale of the intermediate feature maps of the teacher model will also increase proportionally. To match the scale of the feature maps for knowledge distillation, we first use simple adaptive pooling to force the scale to match, but the effect is not good. So we proposed the Rescale Block, which contains several upsampling layers with learnable parameters, so that the scale of the feature map of the student model can be enlarged for hierarchical distillation.
>
> $famp_j$: $famp_j$ is used to represent feature maps, which are the objects of hierarchical distillation. Taking ResNet as an example, we output feature maps once after each stage, and calculate the loss of KL divergence after some simple transformations. For example, $famp_1$ represents the intermediate layer representation obtained after stage 1.
>
> ## Reference
> [1] Hankook Lee, SungJu Hwang, and Jinwoo Shin. Self-supervised label augmentation via input transformations. International Conference on Machine Learning, International Conference on MachineLearning, Jul 2020.

---

> ### Author Response · Authors · 2024-11-21
>
> # Q6: Utilize additional linear classification layers for Eq. (3) and Eq. (4)
> Yes, as I replied to Q5-2, we did some simple processing on the intermediate feature layer, including the linear classification layer. Because there is also a loss function calculation between the intermediate feature layer and the true label. Thank you for pointing out that the formula and explanation of this part are a bit imprecise. We will make a clearer explanation in the final version.
> # Q7: Requirements for additional student model experiments
> Variable scale input is proposed for teacher networks with stronger learning capabilities. Through our experiments, we can also find that variable-scale input can make full use of the learning ability of the teacher network. If the variable scale method is used to train students, this is contrary to the concept we proposed. In addition, such training will let the use of Rescale Block also violate the original design motivation, because the teacher network cannot expand its learning capacity to match the student network. Finally, if the student who receives the variable scale input is put into practical application, it needs to receive normal-size input. In this way, the inconsistency between the training size and the test size will also lead to a decrease in the actual performance of the student model. So I think the additional experiments requested by this comment have no reference significance for our work.
> # Q8: Requirements for additional experiments in ablation study
> Thank you for pointing out the areas where our ablation experiment can be improved. Adding some teacher-student pairs can indeed make our ablation experiment conclusions more credible. We will add experiments immediately and the experimental results will be shown in the reply as soon as possible.
> # Q9: Requirements of additional experimental results
> Of course, the following content is a supplement to Table 2 and Table 3.
> Teacher                      | WRN40-2 | ResNet56 | ResNet32x4 | VGG13 | ResNet50 |
> ---------------------------- | --------------|----------------|-----------------|------------|---------------|
> with our method         | 80.49       | 77.29         | 82.87           | 79.28    | 83.66        |
> without our method    | 76.54        | 73.44        | 79.63            | 74.64    | 79.34        |
>
> The main reason for not showing the accuracy of the teacher model in the table is that the teacher model trained by our method receives input of twice the size, while the teacher model without our method receives input of the original size. Even if our teacher model has a higher accuracy, it is unfair to directly compare the accuracy of the teacher model.
> # Q10: Requirements of additional ablation study results
> Our ablation experiments mainly explore different modes of teacher input and different combinations of loss functions. If the teacher model is not trained, these ablation experiments cannot be performed at all.
>
> Or if you mean that you need me to supplement the ablation experiment data under traditional knowledge distillation, the ablation experiment data of transformation is the blue content (baseline kd) in Figure 4(a), and the ablation experiment of loss is meaningless for exploring the loss function under our variable scale distillation framework.
> # Q11-1: Provide comparison results for the teacher model's feature representations
> Thank you for your suggestion. The comparison of the intermediate feature representations of the teacher model with different input sizes can indeed enhance the credibility of our work. However, considering that our experiments are mainly conducted on low-resolution image datasets such as cifar100, the visualization of the intermediate features reflects the advantages of our framework. So after giving up this point, we choose to visualize the correlation matrix of logits to reflect the advantages of variable-scale input.
> # Q11-2: Concerns about Correlation Matrix of Logits
> Thank you for your question. We may not be able to let you understand this part in the text description. We will give a clearer explanation in the final version.
>
> The teacher model and student model obtained by our method have lower correlation in logits under different categories. In Figure 5, the color at [i, j] (i≠j) of the right image (using our method) is lighter than the left one (without our method). This means that under our method, the teacher model transfers less inference information of the wrong category to the student model. These two pictures can prove the advantages of the variable scale input we proposed.
> # Writing Problems
> Thank you for your detailed review of our work and for pointing out some problems in our writing. We apologize if this has caused you a bad reading experience or misunderstanding. At the same time, we will further check all possible writing problems and avoid them in the final version.

---

> ### Author Response · Authors · 2024-11-23
> **Additional experiments of ablation study**
>
> We think the supplement to the ablation experiment you proposed in question 8 is very important, so we added experiments on different transformations and different loss functions. Considering that time is not sufficient, we currently added a pair of teachers and students, the teacher is wrn40-2, and the student is shufflenetv1, which has different architecture from the teacher's.
>
> transformation    | student accuracy
> ---------------------- | -----
> none (baseline)  | 74.83
> 2x & Perm.         | 77.67
> Rot.                    | 80.11
> 4x & Rot.            | 81.76
> 2x & Rot.            | 83.96
>
> loss                     | student accuracy
> -----------------------|----------------------
> combination 1     | 76.21
> combination 2     | 80.43
> combination 3     | 81.57
> combination 4     | 83.96
>
> The loss function combinations represented by combination1, 2, 3, and 4 are in the same order as in Figure 4(a) in our article. The data distribution of the experimental results is consistent with that in Figure 4, indicating that the student model trained by doubling size input and rotation method and the selected loss function performs best. Thank you again for pointing out the possible inaccuracies in our ablation experiment. We will add the above experimental data to the final version.

---

> ### Comment · Reviewer_JDMz · 2024-11-25
>
> Thanks for your responses! Most of my questions have been resolved, and I’d like to increase my rating from 3 to 5.
> Here is why I rate it 5:
>
> (1) Motivation
>
> One of the key advantages of knowledge distillation (KD) in optimization is the ability to improve the performance of the student network without needing to train the computationally expensive teacher network. However, this paper takes the approach of training the teacher, which naturally requires a strong justification.
> Based on the paper and the responses provided (Answer 1 and Answer 2), it seems the authors justify their choice of training the teacher network by arguing that teacher network, with its stronger learning capabilities compared to the student network, is better equipped to effectively transfer information about rotation or scale obtained through self-supervised learning tasks.
> This reasoning implies two key assumptions:
>
> 1.	Student networks are unable to sufficiently learn scale or rotation information on their own through self-supervised learning and therefore require this knowledge to be transferred from a trained teacher network.
>
> 2.	Pre-trained teacher networks are less effective than trained teacher networks in transferring scale or rotation information to student networks.
>
> I believe the paper needs to provide sufficient experiments and explanations to support these two assumptions. These points are related to Question 7 and Question 10, respectively. If a student network can learn this capability directly or if a pre-trained teacher network can effectively transfer it, the need for training the teacher network becomes questionable.
>
> (2) Contribution
>
> The above concerns regarding motivation weaken the contribution of extending self-supervised learning tricks to the KD framework. Beyond this, it is unclear what other contributions the paper offers.

---

> > ### Author Response · Authors · 2024-11-25
> >
> > First of all, thank you very much for your reply and willingness to improve the rating, which is undoubtedly a recognition of our work. Next, I would like to answer the remaining two questions.
> >
> > For the first question, I think the student model can also improve its performance through self-supervised tasks, but if we only train the student without training the teacher, this is not a discussion within the scope of knowledge distillation. Additionally, in our work, in addition to receiving guidance from the teacher, the student model itself also enhances its performance through self-supervised task such as rotation. Therefore, in our work, the student model not only learns the knowledge of the teacher model in the rotation task, but also enhances its own performance through the self-supervision task.
> >
> > For the second question, the pre-trained model is trained with the original scale, so if we want to transfer the knowledge brought by the variable scale input, we can only train a new variable-scale teacher instead of using the pre-trained teacher. As for the knowledge transfer of rotation, it is discussed in [1], which is cited in our work.
> >
> > I'm sorry that I may not have expressed my ideas clearly in the previous reply. I hope this reply can solve the remaining two problems. In short, I think these two problems can be solved by theoretical explanations and experiments in existing work, so we did not add additional experiments. Finally, thank you again for your willingness to review our work, read my reply and improve the rating.
> >
> > ## Reference
> > [1] Chuanguang Yang, Zhulin An, Linhang Cai, and Yongjun Xu. Hierarchical self-supervised aug-mented knowledge distillation. In Proceedings of the Thirtieth International Joint Conference on Artificial Intelligence, Aug 2021. doi: 10.24963/ijcai.2021/168. URL http://dx.doi.org/10.24963/ijcai.2021/168.

---

> > ### Author Response · Authors · 2024-11-30
> > **Additional experiment for remaining questions**
> >
> > We fully respect your views on the remaining two questions. Although we have explained them to some extent in the previous reply, we believe that additional experiments can make our arguments more convincing. So we conducted some experiments to support our reply.
> > # Q1: student needs knowledge transferred from the teacher
> > We apply the rescaling input and rotation directly to the student instead of using the teacher model to pass this information. The experimental results are as follows.
> >
> > student                                                               | accuracy
> > --------------------------------------------------------------|--------------
> > wrn40-1 (without knowledge from the teacher) | 73.28
> > wrn40-1 (with knowledge from wrn40-2)           | 79.27
> > wrn16-2 (without knowledge from the teacher) | 73.59
> > wrn16-2 (with knowledge from wrn40-2)           | 81.82
> >
> > Due to time constraints, we only tested these two students. The results showed that students’ learning effects were not as good as those imparted by teachers. We hope this result will make our response more convincing, and we will also add this in the final version.
> >
> > # Q2: pre-trained teacher networks are less effective
> > When we do not train the teacher by variable scale input and rotation, but use the pre-trained teacher, we obtain the following results.
> >
> > student                                                               | accuracy
> > --------------------------------------------------------------|--------------
> > wrn40-1 (from pre-trained wrn40-2)                  | 77.00
> > wrn40-1 (our method)                                        | 79.27
> > wrn16-2 (without knowledge from the teacher) | 77.20
> > wrn16-2 (with knowledge from wrn40-2)           | 81.82
> >
> > Experimental results show that students taught by teachers trained by our method outperform students taught by pre-trained teachers. I think this experiment can better illustrate the necessity of training a new teacher, and we will add corresponding text descriptions in the final version. Thank you again for your suggestions on our experiment.

---

> ### Comment · Reviewer_JDMz · 2024-12-01
>
> Thank you for addressing my two concerns. While your additional experiments were conducted using a single pair of networks, which limits generalizability, I can see that receiving scaled image information from a trained teacher is effective. Extending this approach to other networks in future work could strengthen your findings. However, despite these contributions, I would like to maintain my current score for the following reasons:
>
> >### 1. Academic Integrity
>
> In response to my previous question, you referenced HSAKD and stated that you provided a citation.  After reviewing the HSAKD paper, I found their method to be strikingly similar to the loss terms you described in Sec. 3.2.2(i.e., $L^S_{agg1}$, $L^S_{agg2}$, $L^S_{KD1}$, and $L^S_{KD2}$). Additionally, as you acknowledged in Q6, you employed an additional linear classification step, which appears to be borrowed from the task loss and mimicry loss framework of HSAKD. Furthermore, the ablation study shown in Fig. 3 (left) of HSAKD closely resembles the ablation study in Fig. 4(b) of your work.
>
> While borrowing ideas from prior work is not inherently problematic, the lack of clear explanation and citation in the Training section(Sec.3.2.2) of the Method, as well as the absence of explicit acknowledgment in the Experiment section, creates a significant risk of confusing readers. Without clarification, readers may mistakenly believe these methodologies are entirely novel to your work. This is a serious concern. If the four loss terms in your method differ from HSAKD’s, I urge you to clearly explain these differences in detail.
>
> >### 2. Novelty
>
> Although the initial similarity caused some confusion, I understand that the novelty of your work lies in two main aspects as described in your paper:
>
> - Adding scale transformations to HSAKD.
> - Introducing a Rescale block.
>
> However, these contributions feel somewhat incremental, and I remain unconvinced of their broader impact. Moreover, the lack of detailed explanation about the Rescale block and the absence of ablation studies related to it (even after the revision period) leave me questioning the novelty and effectiveness of this method.
>
> >### 3. Clarity
>
> Despite the discussion phase and additional feedback provided, none of the reviewers’ concerns appear to have been addressed in the revised manuscript. While your responses in the OpenReview comments provide some clarification, the manuscript itself remains unchanged. This makes it difficult to evaluate whether the concerns have been resolved in a meaningful way. Additionally, the claims in the manuscript remain unclear.
>
> While I acknowledge the effort you’ve put into addressing the concerns during the discussion, the above issues make it difficult for me to change my score at this time.

---

### Official Review · Reviewer_BeDJ · 2024-11-03

**Soundness:** 2
**Presentation:** 2
**Contribution:** 2
**Rating:** 5
**Confidence:** 4

**Summary:**

This paper proposes the Variable Scale Distillation (VSD) Framework for knowledge distillation, where a teacher network processes images at varying scales to capture richer feature representations. A Rescale Block is introduced to maintain scale consistency between teacher and student features, aiming to improve the knowledge transfer process. Experimental results on different datasets demonstrate improved student performance compared to existing distillation methods.  However, I have several concerns, especially regarding the novelty, evaluation rigor, and justification of the proposed Variable Scale Distillation (VSD) framework.

**Strengths:**

1. Leveraging multi-scale images in the teacher network could potentially enrich the knowledge transferred to the student which is interesting.
2. This paper opens up the concept of utilising teacher potential.
3. This addition aims to maintain feature consistency, addressing common misalignment issues in hierarchical knowledge transfer.

**Weaknesses:**

1. Both variable scaling and self-supervised integration have been explored separately in past work, and combining them here lacks a distinct methodological contribution.
2. There’s no clear analysis explaining why rescaling improves transfer quality. Empirical results are shown, but insights or qualitative explanations are missing.
3. The paper does not explore how different scaling techniques might impact results, which could limit the framework’s robustness.
4. The framework is tested mainly on CIFAR100 (lower resolution), which raises questions about its generalizability. Additional tests on larger, varied datasets would add credibility. It is suggested to evaluate the proposed approach on higher-resolution based images.

**Questions:**

1. Please discuss how this framework differs conceptually and practically from existing methods using multi-scale and self-supervised learning in distillation.

2. How does the choice of scaling method (e.g., bilinear interpolation) affect the results? Please consider testing different scaling methods to assess the robustness of the framework, as certain methods may impact performance differently.

3. Why rescaling improves feature transfer? Is it possible to provide any theoretical or visual analysis that supports this? Including feature-level visualizations or theoretical justifications would help clarify why rescaling benefits knowledge transfer.

4. Is it better to test the VSD framework on larger and more diverse datasets of higher resolution? Expanding evaluation to larger datasets would help validate the generalizability and broader applicability of the framework.

5. It is suggested to conduct ablation studies to isolate the impact of the Rescale Block on the student’s performance.

---

> ### Author Response · Authors · 2024-11-21
> **Rebuttal for reviewer BeDJ**
>
> Thank you for acknowledging our work, including rescaling the input, introducing the rescale block, proposing a novel refinement framework, and considering our work valuable in advancing research, which is very encouraging to us. Thank you again for your detailed comments, encouragement, and recognition.
> # Q1: Combination of variable scaling and self-supervised (weakness 1)
> Variable scale and self-supervised tasks have indeed been explored separately in the field of knowledge distillation. The reason why we combine them into our proposed distillation framework is that variable scale of input can increase the amount of information contained in the same image, and self-supervised tasks such as rotation can increase the number of samples for the model to learn. Both of these points can make full use of the learning ability of the teacher model, which is also one of the innovations of this work.
> # Q2: Why rescaling improves feature transfer (weakness 2 & question 3)
> In Figure 5 of the article, we try to use visualization to explain why variable scale input can improve model performance. When using variable scale input (our method), the correlation between the teacher model logits and the student model logits in different categories is lower, which is reflected in the lighter color of the visualization image compared with the another one. This shows that compared with the original scale input, our method learns less reference about the wrong category in the process of students learning from the teacher.
>
> As for visualizing the intermediate layer features, since our experiments are mainly based on the cifar100 dataset, the visualization effect of the intermediate layer features of low-resolution images is not good, so we did not consider providing theoretical proof from this aspect.
> # Q3: Choice of scaling method (weakness 3 & question 2)
> Thank you for pointing out the shortcomings of our experiment. First of all, bilinear interpolation is chosen as the rescaling method because it is the most one of the most commonly used interpolation method. In addition, bilinear interpolation is only used to increase the image resolution. The focus of our distillation framework is that high-resolution images can fully utilize the learning ability of the teacher model, so we did not discuss the impact of different interpolation methods on model performance.
>
> However, considering the rigor of the experiment, we think your suggestion is very inspiring. We will add experiments and update the experimental results as soon as possible.
> # Q4: Requirements for additional experiments (weakness 4 & question 4)
> To increase the credibility of our work, we have conducted experiments on the ImageNet-1k dataset. The experimental results will be updated as soon as possible. Referring to the experimental contents in [1] and [2], we will train the teacher-student model of resnet34 and resnet18 on ImageNet-1k.
> # Q5: How our framework differs conceptually and practically from existing methods (question 1)
> Conceptually, our method innovatively introduces a variable-scale training approach for the teacher model, which increases the input size by a factor of two. This allows the teacher model to learn from a wider range of feature scales, providing richer representations that can be more effectively distilled into the student model. Unlike traditional self-supervised distillation methods, our approach combines the self-supervision task with image scaling to maximize the exploration of the learning capabilities of the teacher model.
>
> Practically, we propose the Rescale Block, which adjusts the teacher's intermediate layer sizes to align with the student model’s features. This block introduces additional learnable parameters, allowing the teacher model to adapt to the scaling process and facilitate more efficient knowledge transfer. By enhancing the teacher's capacity to capture multi-scale features, we are able to improve the overall distillation performance in a way that existing methods using only single-scale or standard self-supervised tasks do not. This novel combination of variable scale input processing and self-supervised tasks sets our framework apart.
>
>
>
> ## References:
> [1] Guodong Xu, Ziwei Liu, Xiaoxiao Li, and Chen Change Loy. Knowledge Distillation Meets Self-Supervision, pp. 588–604. Jan 2020. doi: 10.1007/978-3-030-58545-7 34. URL http://dx.doi.org/10.1007/978-3-030-58545-7_34.
>
> [2] Chuanguang Yang, Zhulin An, Linhang Cai, and Yongjun Xu. Hierarchical self-supervised aug-mented knowledge distillation. In Proceedings of the Thirtieth International Joint Conference on Artificial Intelligence, Aug 2021. doi: 10.24963/ijcai.2021/168. URL http://dx.doi.org/10.24963/ijcai.2021/168.

---

> ### Author Response · Authors · 2024-11-21
>
> # Q6: Concerns about the impact of Rescale Block (question 5)
> Thank you for your suggestion. We have previously conducted experiments before proposing the Rescale Block, we used a simpler adaptive pooling to match the intermediate feature sizes between the teacher and student models. However, this approach did not produce satisfactory results. So we proposed the Rescale Block, which is crucial for aligning feature scales more effectively. We believe that this comparison can serve as an ablation study of the performance of the rescale block on the student. Of course, we will illustrate this in the final version to emphasize the role of the Rescale Block in our distillation framework.

---

> ### Author Response · Authors · 2024-11-23
> **Additional experiment of different scaling method**
>
> We think the additional scaling method experiment you proposed in question 2 is very necessary. We added two scaling methods based on our work, namely nearest and bicubic. The following are the experimental results
>
>  method     | wrn 40-2 (teacher)    | wrn 16-2 (student)
> --------------- | ----------------------------|-------------------------
> nearest       | 79.64                        | 79.42
> bicubic        | 78.68                        | 79.57
>
> Among the three existing scaling methods, the bilinear interpolation selected in our article has the greatest improvement in student model performance. However, the student model performance in our supplementary experiments also exceeds the previous distillation method. This also shows that the advantage of our method lies in the variable scale input itself, and the scaling method is not the main focus of the distillation framework. The additional experiments have increased the rigor of our work, and we will still add instructions on the scaling method in the final version.

---

> ### Author Response · Authors · 2024-11-30
> **Additional experiment on ImageNet**
>
> We started the experiment on the ImageNet dataset as soon as we learned about the reviewer's comments, and the results are as follows.
>
> Accuracy                 | Top - 1   | Top - 5
> ---------------------------|------------|----------
> ResNet34 (teacher) | 78.01    | 71.50
> ResNet18 (student) | 93.89    | 90.50
>
> Our teacher model surpasses all teachers obtained by current distillation methods, and the students reach the sota level, surpassing the students obtained by most methods. In view of this experimental result, we believe that our method is suitable for low-resolution images. In our existing experiments, the performance on the Cifar100 and tinyImageNet datasets far exceeds the existing methods. And when the image resolution increases, it is not so necessary to increase the resolution so that the teacher can learn more. This is why our current method surpasses most other methods but not all.

---

> > ### Comment · Reviewer_BeDJ · 2024-12-02
> > **Response to the authors**
> >
> > Thanks to the authors for the reply. Even, the authors answered all of my concerns. I choose to keep my score.
> >
> > Justification of my rating:
> >
> > 1. It is not still clear "Why rescaling improves feature transfer (weakness 2 & question 3)" in the responses. I feel the discussion both here and in the manuscript is not enough to justify. As this is the heart of the paper, in my opinion, this discussion should be more extensive with theoretical proof.
> >
> > 2. Additionally, Some of the results raise confusion (ImageNet table of Reviewer nq3Z).
> >
> > 3. In my opinion, I feel this paper has potential but needs to be improved a lot with careful concentration.

---

### Official Review · Reviewer_nq3Z · 2024-11-04

**Soundness:** 2
**Presentation:** 2
**Contribution:** 2
**Rating:** 5
**Confidence:** 4

**Summary:**

The paper introduces the Variable Scale Distillation Framework (VSDF), which enhances teacher-student knowledge transfer by training the teacher on inputs of varying scales. This approach is supported by a Rescale Block that aligns feature maps between teacher and student, addressing the feature mismatch issue caused by differences in input scale. Additionally, a self-supervised aggregated task is incorporated to improve the teacher’s feature extraction capacity, enabling the student network to achieve better generalization and accuracy, especially in low-data scenarios. The authors demonstrate this approach’s efficacy on CIFAR-100, showing improvements in both classification accuracy and transfer learning.

**Strengths:**

1. The VSDF enables the teacher network to leverage richer features by introducing variable scale inputs, which pushes beyond typical one-to-one input matching in distillation.

2. This framework taps into the teacher’s full capacity and could offer substantial improvements in cases where the student network benefits from a diverse set of learned representations.

3. The Rescale Block’s design addresses a common challenge in knowledge distillation: mismatched feature scales between teacher and student networks.

4. The paper provides a well-rounded empirical analysis with an ablation study, comparisons to state-of-the-art distillation methods, and few-shot learning performance.

**Weaknesses:**

1. The introduction of VSDF and Rescale Block raises concerns about computational overhead. The experiments would benefit from additional metrics on training time, computational cost, or memory usage compared to standard distillation techniques, which could affect the method’s practicality in resource-limited environments.

2. While the teacher benefits from richer inputs, this may not always yield proportionally significant improvements in the student’s performance, especially when dealing with datasets of smaller resolutions or non-visual domains where variable scale inputs may not be as effective.

3. CIFAR-100 (32X32) is a commonly used dataset, but the lack of experiments on larger or more complex datasets with higher resolution limits the generalizability of the results.

4. While the self-supervised aggregated task shows promising results, the paper does not sufficiently analyze the contribution of individual self-supervised tasks (such as rotation versus permutation) to overall performance.

5. this approach might benefit from a clearer theoretical foundation or explanation, particularly regarding why specific transformations, like rotation, improve generalization more than others. Without this, the choice of transformations may appear somewhat arbitrary.

6. The hierarchical distillation losses and the aggregated task losses are complex and potentially difficult to tune. This complexity may limit reproducibility or make the method sensitive to hyperparameter selection. A more streamlined loss function could simplify the application of this framework without sacrificing performance.

**Questions:**

1. What is the added computational cost of VSDF and the Rescale Block? Please evaluate the training time complexity.

2. Is it possible to evaluate VSDF on larger datasets like ImageNet?

3. What is the impact of each self-supervised task (e.g., rotation vs. channel permutation)? Why might some perform better? Please explain.

4. How does VSDF scale with larger teacher or student models? Are there any scenarios this method struggles?

5. Please provide details of the hyperparameters tunning for the loss functions.

6. Does the Rescale Block work effectively across different input resolutions or tasks? It is better to explain and evaluate explicitly.

---

> ### Author Response · Authors · 2024-11-21
> **Rebuttal for reviewer nq3Z**
>
> Thank you for acknowledging our work, including rescaling the input, introducing the rescale block, proposing a novel refinement framework, and considering our work valuable in advancing research, which is very encouraging to us. Thank you again for your detailed comments, encouragement, and recognition.
> # Q1: Concerns about added computational cost (weakness 1 & question 1)
> In our work, we mainly proposed "variable input scale" and "rescale block", among which variable input scale is the factor that increases the time complexity of this work. When training a pair of teacher and student, only the training time of the teacher model increased by about 3 times, while the training time of the student network was comparable to the baseline model.
>
> For the practicality of our framework, the additional time cost does affect the training time. However, choosing the mixed precision training mode in actual training can also reduce the training cost to a certain extent. Considering the final performance improvement, we still believe that this framework has certain practicality.
> # Q2: Ineffectiveness of variable scale input on datasets of smaller resolutions (weakness 2)
> In the experiments we have done so far, we have covered several common public datasets such as cifar100, stl10 and tinyimagenet. The first two datasets have lower resolutions. We have demonstrated the sota performance of our method in related experiments.
> # Q3: Requirements for additional experiments (weakness 3 & question 2)
> We are conducting additional experiments on imagenet and the experimental results will be updated as soon as possible. Referring to the experimental contents in [1] and [2], we will train the teacher-student model of resnet34 and resnet18 on Imagenet-1k.
> # Q4: Contribution of individual self-supervised tasks (weakness 4)
> The idea of ​​self-supervised aggregation task comes from [3]. The fundamental reason for using this method is to make full use of the more powerful learning ability of the teacher model. Analyzing the contribution of aggregation task to performance can indeed highlight the innovation of variable scale in the framework and the rigor of the article. We will add relevant instructions in the final version.
> # Q5: Choice of rotation as self-supervised transformation (weakness 5)
> The selection of self-supervised tasks refers to [3]. The article compares the improvement of permutation and rotation compared with the baseline through experiments. The results show that rotation has the best performance improvement for the model. We apologize for not explaining this point clearly in the text. We will explain this in the final version.
> # Q6: Hyperparameters tunning for the loss functions (weakness 6 & question 5)
> The hyperparameters for the loss functions were kept simple, with each loss component set to a weight of 1. After referring to [1] and [2], which are also distillation work on self-supervised tasks, we found that the weights of each item in their loss function are all 1, that is, all components contribute the same proportion to the training. Thus, we did not perform additional hyperparameter tuning for the loss functions in our work.
> # Q7:  Impact of each self-supervised task (question 3)
> We appreciate your interest in the impact of the self-supervised tasks. We believe that teacher's inaccurate self-supervision output encodes rich structured knowledge of teacher network and mimicking this output can benefit student's learning [1]. When the self-supervision task is aggregated with the classification task into a new task, we believe that this can further explore the learning ability of the model and provide a simple end-to-end training method.
>
> As for the reasons behind the differing performances of these tasks, it seems that rotation prediction typically yields better results in practival [3] because it directly leverages spatial information, making it more aligned with the underlying structure of the data. In contrast, channel permutation is a more abstract transformation and may not be as directly useful for capturing spatial relationships.
>
> ## References:
>
> [1] Guodong Xu, Ziwei Liu, Xiaoxiao Li, and Chen Change Loy. Knowledge Distillation Meets Self-Supervision, pp. 588–604. Jan 2020. doi: 10.1007/978-3-030-58545-7 34. URL http://dx.doi.org/10.1007/978-3-030-58545-7_34.
>
> [2] Chuanguang Yang, Zhulin An, Linhang Cai, and Yongjun Xu. Hierarchical self-supervised aug-mented knowledge distillation. In Proceedings of the Thirtieth International Joint Conference on Artificial Intelligence, Aug 2021. doi: 10.24963/ijcai.2021/168. URL http://dx.doi.org/10.24963/ijcai.2021/168.
>
> [3] Hankook Lee, SungJu Hwang, and Jinwoo Shin. Self-supervised label augmentation via input transformations. International Conference on Machine Learning,International Conference on MachineLearning, Jul 2020.

---

> ### Author Response · Authors · 2024-11-21
>
> # Q8: How does VSDF scale with larger teacher or student models (question 4)
> Although we have not yet conducted experiments with significantly larger teacher or student models, we believe that this approach should be able to be effectively adapted to larger teacher and student models. The Rescale Block itself is designed to be modular and flexible, and we expect it to work well across a range of model sizes.
> # Q9: details about Rescale Block (question 6)
> The structure of the Rescale Block adapts based on the rescale factor, which is adjusted to accommodate different input sizes. While we initially experimented with adaptive pooling to handle varying resolutions, we found that this approach did not yield satisfactory results because spatial information is lost when forcing uniform size through adaptive pooling. This is why we propose Rescale Block instead of using simple adaptive pooling to force a uniform size of feature layers.

---

> ### Comment · Reviewer_nq3Z · 2024-11-29
> **Reply to the Authors**
>
> Dear Authors, Thank you for the reply. There some concerns that are not addressed properly:
>
> Q1. Is it possible to measure training complexity quantitatively?
>
> Q2. Still there is no evaluation if this method is effective for large-scale dataset or higher resolution. If the basic resolution of the image is $226 \times 226$, then the images are up-scaled into higher resolution to feed to the teacher which raises two concerns: 1) training time and complexity and 2). The performance of the teacher model is highly depended on the performance of the scaling method
>
> Q3. The authors uses 1 as the balancing weights for every loss term. Having experimental evaluation for different values of hyper-parameters or balancing weights is very crucial for further analysis. Without having these analysis it is difficult to understand the effect of the proposed approach. Also the scale and distribution of different losses is different. It does not make sense to use 1 to all the loss terms without performing evaluation.

---

> > ### Author Response · Authors · 2024-11-30
> > **Rebuttal for reviewer nq3Z**
> >
> > Thank you for reading our response. We will further answer the remaining three questions in this reply.
> > # Q1: measure training complexity quantitatively
> > Of course, we checked the log files of model training. For the wrn40-2 network (teacher), the training time for one epoch increased by 4.71 times. For the resnet32x4 network (teacher), the training time for one epoch increased by 3.13 times. For the wrn16-2 network (student), the training time for one epoch remained basically unchanged (1.0002 times). The above data is the multiple relationship obtained by randomly selecting 10 epochs. For a small-scale teacher such as wrn40-2, the increase is more obvious; for a larger teacher such as resnet32x4, the increase is relatively small. As for the student model, we can assume that there is no increase in time complexity.
> > # Q2: experiments on ImageNet
> > We started the experiment on the ImageNet dataset as soon as we learned about the reviewer's comments, and the results are as follows.
> >
> > Accuracy                 | Top - 1   | Top - 5
> > ---------------------------|------------|----------
> > ResNet34 (teacher) | 78.01    | 71.50
> > ResNet18 (student) | 93.89    | 90.50
> >
> > Our teacher model surpasses all teachers obtained by current distillation methods, and the students reach the sota level, surpassing the students obtained by most methods. In view of this experimental result, we believe that our method is suitable for low-resolution images. In our existing experiments, the performance on the Cifar100 and tinyImageNet datasets far exceeds the existing methods. And when the image resolution increases, it is not so necessary to increase the resolution so that the teacher can learn more. This is why our current method surpasses most other methods but not all.
> > # Q3: hyper-parameters of loss function
> > First of all, thank you very much for pointing out the possible inaccuracies in our work and other related work. We did not consider modifying the weights of each item in the loss function. We immediately supplemented the experiment. In order to simplify the complexity of the experiment, we divided the loss function into classification loss and distillation loss (KL divergence). Through 10 random searches, we determined the best-performing weight ratio. The classification loss weight is 0.758, and the distillation loss weight is 0.845. Although this ratio is close to 1, the student accuracy trained at this ratio is higher. So we will explain this in the final version. At the same time, thank you again for pointing out the shortcomings of our work and related work.

---

> > > ### Comment · Reviewer_nq3Z · 2024-12-02
> > > **Response to the Authors**
> > >
> > > Dear authors,
> > >
> > > Thank you for the responses. I appreciate the effort. The authors responded some of my concerns perfectly where some of the concerns are not well addressed.
> > >
> > > 1. The results in ImageNet table looks wired. The teacher network achieves 78.01 as the top-1 accuracy where the top-5 accuracy is 71.50? On the other hand student Top-1 accuracy is 93.89? that much greater than teacher? where also top-5 accuracy is less that top-1. Is there any mistakes putting the results on the table?
> > >
> > > 2. There are no comparison or discussion with state-of-the-art methods.
> > >
> > > 3. Also the analysis is not enough in this version.
> > >
> > > 4. I am also agreed with reviewer JDMz, specifically regarding novelty.
> > >
> > > Though I appreciate author responses and their efforts. I still believe this paper needs more time to be improved. For these reasons I would like to keep my score unchanged.

---

### Meta-Review · Area_Chair_HuPV · 2024-12-19

**Metareview:**

This paper investigates knowledge distillation. This paper proposes a new method that introduces the rescale block, which preserves scale consistency during hierarchical distillation, allowing the teacher to extract richer, more informative features. The experimental results demonstrate the effectiveness of the proposed method.

Pros:

- The perspective of enhancing the teacher's learning process is interesting.
- The introduced rescale block is reasonable to address the feature mismatch problem.

Reasons to reject:

- The biggest concern lies in that the applicability is very limited. It is noteworthy that the key advantage of knowledge distillation is the ability to improve the performance of the student network without the need of training the computationally expensive teacher network. However, this paper will train the teacher model and the student model together, which clearly contradicts the main purpose of knowledge distillation. While improving the teacher's learning process seems interesting, it will actually increase the computational burden, which should be avoided in knowledge distillation.

- The novelty of this work lies in two main aspects, including adding scale transformations to HSAKD and introducing a rescale block, while these contributions seem to be incremental and lack broader impacts.

- There are important flaws on the reported experimental results on ImageNet, where the student model outperforms the teacher model and the top-1 accuracy is better than the top-5 accuracy.

**Additional Comments On Reviewer Discussion:**

This paper finally receives 5, 5, 5, which means, all the reviewers vote for weak rejection.

I have read all the reviews and the authors' rebuttal. The reviewers raised many concerns, among them the most important ones are listed as the above reasons to reject. The authors' rebuttal did not fully convince the reviewers. I agree with the reviewers that this paper needs further to be improved.

---

### Decision · Program_Chairs · 2025-01-22

Reject